# Cor Triatriatum Sinistrum Combined with Changes in Atrial Septum and Right Atrium in a 60-Year-Old Woman

**DOI:** 10.3390/medicina57080777

**Published:** 2021-07-30

**Authors:** Horst Claassen, Christian Busch, Matthias Stefan May, Martin Schicht, Michael Scholz, Marko Schulze, Friedrich Paulsen, Andreas Wree

**Affiliations:** 1Department of Anatomy, Rostock University Medical Center, Gertrudenstrasse 9, 18057 Rostock, Germany; marko.schulze@uni-bielefeld.de (M.S.); andreas.wree@med.uni-rostock.de (A.W.); 2Institute of Functional and Clinical Anatomy, Friedrich-Alexander-University Erlangen-Nürnberg, Universitätsstrasse 19, 91054 Erlangen, Germany; martin.schicht@fau.de (M.S.); michael.scholz@fau.de (M.S.); friedrich.paulsen@fau.de (F.P.); 3Department of Internal Medicine, Federal Armed Forces Hospital, Lesserstrasse 180, 22049 Hamburg, Germany; christian1busch@bundeswehr.org; 4Department of Radiology, University Hospital Erlangen, Friedrich-Alexander-University Erlangen-Nürnberg, Maximiliansplatz 3, 91054 Erlangen, Germany; matthias.may@uk-erlangen.de; 5Imaging Science Institute Erlangen, Ulmenweg 18, 91054 Erlangen, Germany; 6AG 3 Anatomie und Zellbiologie, Medizinische Fakultät OWL Universität Bielefeld, Morgenbreede 1, 33615 Bielefeld, Germany; 7Department of Topographic Anatomy and Operative Surgery, Sechenov University, Rossolimo Street 15/13, 119992 Moscow, Russia

**Keywords:** adult, heart atria, abnormalities, cor triatriatum sinistrum, intra-atrial membrane, measurements, atrial wall thickness, ventricular wall thickness

## Abstract

*Background and Objectives*: A rare case of cor triatriatum sinistrum in combination with anomalies in the atrial septum and in the right atrium of a 60-year-old female body donor is described here. *Materials and Methods*: In addition to classical dissection, ultrasound and magnetic resonance imaging, computer tomography and cinematic rendering were performed. In a reference series of 59 regularly formed hearts (33 men, 26 women), we looked for features in the left and right atrium or atrial septum. In addition, we measured the atrial and ventricular wall thickness in 15 regularly formed hearts (7 men, 8 women). *Results*: In the case described, the left atrium was partly divided into two chambers by an intra-atrial membrane penetrated by two small openings. The 2.5 cm-high membrane originated in the upper level of the oval fossa and left an opening of about 4 cm in diameter. Apparently, the membrane did not lead to a functionally significant flow obstruction due to the broad intra-atrial communication between the proximal and distal chamber of the left atrium. In concordance with this fact, left atrial wall thickness was not elevated in the cor triatriatum sinistrum when compared with 15 regularly formed hearts. In addition, two further anomalies were found: 1. the oval fossa was deepened and arched in the direction of the left atrium; 2. the right atrium showed a membrane-like structure at its posterior and lateral walls, which began at the lower edge of the oval fossa. It probably corresponds to a strongly developed eustachian valve (valve of the inferior vena cava). *Conclusions*: The case described suggests that malformations in the development of the atrial septum and in the regression of the valve of the right sinus vein are involved in the pathogenesis of cor triatriatum sinistrum.

## 1. Introduction

Cor triatriatum sinistrum is a rare congenital heart malformation characterized by a fibromuscular membrane that divides the left atrium into two chambers of different sizes. It was first described in a postmortem investigation by Church in 1868 (quoted from [1]). The congenital anomaly occurs in 0.4% of patients with congenital heart disease and is found in less than 0.1% of clinically diagnosed cardiopathies [2]. In 1941, Pfennig described a case of cor triatriatum in a 5-week-old infant. Additionally, he was able to report a total of 14 similar cases [3].

Surprisingly, the cor triatriatum is not described in classical embryological textbooks [4,5] or handbooks focusing on heart anatomy [6]. Even textbooks of internal medicine do not mention this anomaly [7].

Despite the rarity of a cor triatriatum sinistrum, there has been a significant increase in its diagnosis, mainly due to improved imaging techniques [8,9]. Therefore, the possibility of a left atrial membrane should be considered if there are signs and symptoms of mitral stenosis [8]. The present report describes a cor triatriatum sinistrum in a 60-year-old female body donor that was combined with additional anomalies in the right atrium and atrial septum.

## 2. Materials and Methods

A cor triatriatum was found in a 60-year-old female body donor during the gross anatomy course of the Institute of Anatomy and Cell Biology of Martin-Luther-University Halle-Wittenberg. The body donor was 60 years old when she passed away due to carcinoma of the esophagus combined with tumor-related cachexia and chronic bronchitis. Besides classical (heart) dissection, ultrasound (General Electric Vivid E9), magnetic resonance imaging (Siemens Skyra, 3-Tesla), computed tomography (Somatom force, Siemens Healthcare GmbH, Forchheim, Germany) and cinematic rendering were performed. Photographs were taken and the variations were illustrated by a professional institutional illustrator. A reference series of 59 regularly formed hearts (33 males between 51 and 99 years of age, 26 females between 57 and 97 years of age) was used to analyze peculiarities of the left and right atrium together with the atrial septum. None of these hearts presented macroscopically visible pathological alterations. This series comprised of hearts from dissection courses in the summer terms 2016 (22 hearts), 2017 (21 hearts) and 2018 (16 hearts) at the Institute of Anatomy of the Rostock University Medical Center. By investigating quite regularly formed hearts we intended to analyze the possible pathogenesis of the membrane partly dividing the left atrium. In addition, atrial and ventricular wall thickness was measured in the variant heart and in regularly formed hearts of 7 males and 8 females from the dissection course in the summer term 2021 at the Institute of Anatomy of the Rostock University Medical Center. Measurements were taken using a caliper. For a better visualization of the morphological characteristics of the cor triatriatum, a video sequence was created. Cinematic rendering technology was used, which allows a photorealistic representation of the organ and its structures based on the volumetric data of the computed tomography scan [10,11,12].

## 3. Results

### 3.1. Macroscopical Analysis

The cor triatriatum described here was a partial division of the left atrium into two chambers by an intra-atrial membrane. The membrane originated in the upper plane of the oval foramen (Figure 1a,b) and was situated 1 cm above the opening of the left atrial appendage. It was pierced by two small openings of 2 × 3 mm and 3 × 5 mm, respectively. By comparing the heights of the lower and upper parts of the divided left atrium, a proportion of 2:1 was measured. All four pulmonary veins flowed into the upper part of the left atrium. Dissection of the lungs including the pulmonary veins did not reveal any obvious pathological alterations. Since the membrane measured approximately one quarter of the left atrium in cross section, the connection between the two parts of the atrium was broad. In addition, the right atrium contained a membrane-like structure at the posterior and lateral wall (Figure 1c,d). This membrane started at the lower rim of the oval fossa. The membrane suggested a strongly developed valve of the inferior vena cava (eustachian valve). The oval fossa was deeper compared with normal hearts (Figure 1c,d). However, no opening was visible in the oval fossa in the direction of the left atrium.

### 3.2. Measurements of Atrial and Ventricular Wall Thickness

Atrial wall thickness in cor triatriatum sinistrum was 1.00 mm on the right and 2.00 mm on the left. Further on, ventricular wall thickness was 3.00 mm on the right and 17.00 mm on the left. For comparison, atrial and ventricular wall thickness were measured in 15 regularly formed hearts (Table 1). The mean atrial wall thickness in seven male hearts was 2.43 mm on the right and 1.71 mm on the left, while the ventricular mean was 6.00 mm on the right and 14.57 mm on the left. Mean atrial wall thickness in eight female hearts was 1.75 mm on the right and 1.38 mm on the left, while the ventricular mean was 5.37 mm on the right and 12.88 mm on the left.

### 3.3. Analysis by Ultrasound, Magnetic Resonance Imaging, Computed Tomography and Cinematic Rendering

In comparison with a quite normal heart (Figure 2a,c), ultrasound and magnetic resonance imaging clearly showed the membrane of the left atrium within the variant heart (Figure 2b,d). Computed tomography in combination with grayscale images demonstrated a mineralization of the circumflex branch of left coronary artery of the variant heart (Figure 2e). The three-dimensional reconstruction of a virtual atriotomy revealed an additional mineralization in the right coronary artery and its posterior interventricular branch (Figure 2f). For a dynamic visualization, the special features of the cor triatriatum sinistrum are summarized in a slow-motion video which was added as Appendix A. It is available online at https://www.mdpi.com/article/10.3390/medicina57080777/s1.

### 3.4. Comparison with Left and Right Atria of Regularly Formed Hearts

The observed variations in the left and right atria of a reference series of 59 quite regularly formed hearts are summarized in Table 2. At first, we analyzed the surface of the atrial septum seen from the left atrium. In 59 normal hearts, a smooth atrial septum was found in 44%. In the remaining 56% we observed special structures on or in the atrial septum. The concavity of sickle-like structures mostly opened in a dorsal direction (33.9%), rarely in a cranial (1.7%) or caudal direction (1.7%). Sieve-like depressions in the ventro-cranial part of the atrial septum were observed in 6.8%. Moreover, the bottom of the fossa ovalis was bulged in the direction of the left atrium in 11.9%.

Secondly, we analyzed the surface of the atrial septum seen from the right atrium. In the 58 regularly formed hearts, the oval fossa was hardly pronounced in 12.0%. It was flat in 48.3% and deep in 39.7%. In addition, we searched for further possible special features in the right atrium. In the 58 regularly formed hearts, the eustachian valve was missing in 25.4%. In 49.2% it was normally expressed, and in 25.4% strongly expressed. The Thebesian valve was missing in 27.6%. In 55.2% it was normally expressed, and in 17.2% strongly expressed.

## 4. Discussion

Slight et al. [8] stated that there had been a notable increase in the diagnosis of a cor triatriatum sinistrum due to improved diagnostic tools. Concerning males, one case was found in a 35-week-old male fetus [13]. In adulthood there were nine case descriptions of male patients at ages between 24 and 49 years (cases at 24, 32, 35, 36, 3 × 43, 48 and 49 years of age) [14,15,16,17,18]. The earliest cor triatriatum in females was found in a 3-year-old girl, followed by cases of 14- and 18-year-old girls [17,19,20]. In adulthood the data ranged from 30 to 75 years (nine cases at 30, 31, 39, 45, 49, 62, 55, 58 and 75 years of age) [17,21,22,23,24,25,26]. Taken together, 9 cases were found in males and 12 cases in females. Therefore, the statement of Lima and coworkers [17], that a cor triatriatum sinistrum occurs more frequently in males than in females (1.4:1), is not proven by the literature.

If the connection between the two chambers in a cor triatriatum sinistrum is “too small”, 75% of cases die in infancy [27]. Details of the openings in the obstruction membrane were given in four male patients. Chen and colleagues [15] reported two cases at 35 and 48 years of age with openings in the left atrial membrane, each 2 mm in diameter. In a 24-year-old, two communication openings of 4 mm and 5 mm were described [14]. The obstructive membrane in a 36-year-old had an opening of 1 cm in diameter [15]. Eichholz and colleagues [18] described a multiple-windowed membrane of 4.3 cm in diameter attached to the atrial wall between the openings of the pulmonary veins and the left atrial slope in a 43-year-old patient presenting with recurrent syncope.

Similar descriptions were found in six female cases. The smallest opening of 3 mm in diameter was seen in a 39-year-old woman [26]. In two female patients aged 18 and 58 years, the obstructing membrane had a diameter of 1 cm [20,26]. The broadest openings, at 1.2 cm^2^ and 3.2 cm^2^, were seen in a 30- and 31-year-old, respectively [21,22]. Interestingly, in a 55-year-old with a cor triatriatum sinistrum, intra-atrial communication was broad and no significant transmembrane flow obstruction between the proximal and distal chambers was observed [24].

The case of the 60-year-old woman reported here resembles the 55-year-old female patient described by Park and colleagues [24]. The left atrium presented here showed a membrane with a diameter of 2.5 cm which was penetrated by two small openings of 2 × 3 mm and 3 × 5 mm. The separation membrane left an opening of 4 cm in diameter. On the way to the mitral valve, blood flow could presumably pass unhindered first the upper and then the lower chamber of the left atrium. This may explain why the female body donor described here reached a relatively high age. If the connection between the two chambers of a cor triatriatum sinistrum is too tight, 75% of these patients die in infancy [27]. Unfortunately, there are no clear data available correlating the dimensions of the intra-atrial membrane or the diameter of the remaining communication between the two parts of the left atrium and elaborated functional heart parameters. In the present case, we measured atrial and ventricular wall thicknesses and compared these with the respective parameters in regularly formed hearts (Table 1). Left atrial wall thickness of the cor triatriatum sinistrum amounted to 2.00 mm and was not significantly higher than the mean wall thickness of the left atria of eight female hearts at 1.38 mm.

In the present case, we also observed a strongly developed valve of the inferior vena cava (eustachian valve) in the right atrium. According to Sehra and colleagues [28], the pronounced form of a persistent eustachian valve leads to a cor triatriatum dextrum. Information on the pathogenesis of a cor triatriatum dextrum can be found in the publications of Wyss and colleagues [29] and Malik and colleagues [30]. Normally, the right sinus venosus valve recedes between the 9th and 15th weeks of pregnancy, as the cranial section transforms into the terminal crista and the caudal section into the eustachian and thebesian valves. Any disruption in the regression process can lead to remnants of the valve of the right sinus vein as a single muscle rod, a chiari network or a fenestrated or non-fenestrated membrane called cor triatriatum dextrum [29,30].

A cor triatriatum sinistrum can be associated with other congenital heart defects in up to 80% of newborns and children [2]. These may include atrial and ventricular septal defects. A case of left-sided cor triatriatum in combination with a ventricular septal defect and open arterial duct in a six-month-old infant was described by Cabrera and colleagues [31]. According to Jha and Makhija [32], a common association could be an atrial septal defect seen as ostium secundum or an open foramen ovale. Remarkably, in addition to the membrane in the left atrium, the cor triatriatum sinistrum described here further showed a strongly developed membrane-like eustachian valve in the right atrium and a recessed oval fossa arched towards the left atrium.

With regard to a possible pathogenesis of cor triatriatum sinistrum, four mechanisms have been discussed [1,33]: 1. there is an abnormal growth of the primary septum; 2. the integration of the embryonic, common pulmonary vein into the left atrium is incomplete; 3. the common pulmonary vein is trapped by the left horn of the venous sinus, thus preventing its incorporation into the left atrium; 4. the left superior vena cava persists and this acts on the developing left atrium.

Since the membrane in the present case originates at the upper level of the oval foramen, we assume that abnormal growth of the septum primum [1] may be involved in the pathogenesis of the cor triatriatum sinistrum. Furthermore, the noticeably strong, developed eustachian valve may be due to a failure of the regression of the right valve of the sinus venosus [29,30].

## 5. Conclusions

In the cor triatriatum sinistrum, a membrane in the left atrium was formed at the upper level of the oval foramen, while the prominently developed eustachian valve of the right atrium began at the lower edge of the oval fossa. This leads to the hypothesis that maldevelopments of the atrial septum are involved in the pathogenesis of a cor triatriatum sinistrum. Such malformations will be diagnosed more and more frequently and earlier in life by modern imaging diagnostic techniques. The intra-atrial communication in the cor triatriatum sinistrum of a 60-year-old female body donor described here was wide, and there was no significant flow obstruction between the proximal and distal chambers of the left atrium. However, surgical treatment is indicated in patients with obstructive cor triatriatum sinistrum where the connection between these two chambers is restricted in a way similar to mitral stenosis.

## Figures and Tables

**Figure 1 medicina-57-00777-f001:**
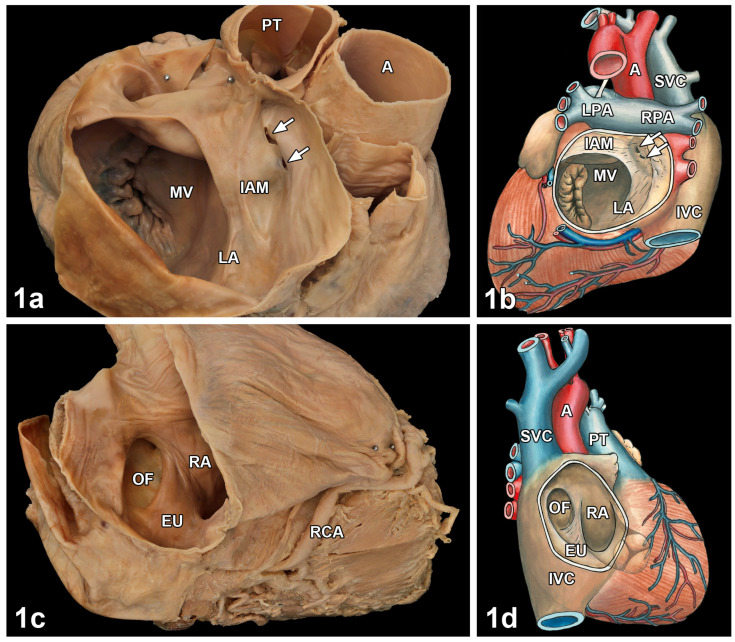
Macroscopic analysis of left and right atria from the heart of a 60-year-old woman. (**1a**) Left atrium (LA) seen from above with pulmonary trunk (PT) and aorta (A). The atrium was divided incompletely in two parts by an intra-atrial septum (IAM). The lower division of the atrium was directed towards the mitral valve (MV), the smaller upper part to the pulmonary veins. Note that the intra-atrial membrane was perforated by two small openings (arrows). (**1b**) Diagram completing the photograph in (**1a**). (**1c**) Right atrium (RA) seen from above with a part of the right coronary artery (RCA). The oval fossa (OF) was deepened. At the dorsal circumference of the atrium, a membrane (EU), taking its origin at the lower rim of the OF was seen. (**1d**) Diagram completing the photograph in (**1c**). A = Aorta, EU = Eustachian valve, IAM = Intra-atrial membrane, IVC = Inferior vena cava, LA = Left atrium, LPA = Left pulmonary artery, MV = Mitral valve, OF = Oval fossa, PT = Pulmonary trunk, RA = Right atrium, RCA = Right coronary artery, RPA = Right pulmonary artery, SVC = Superior vena cava.

**Figure 2 medicina-57-00777-f002:**
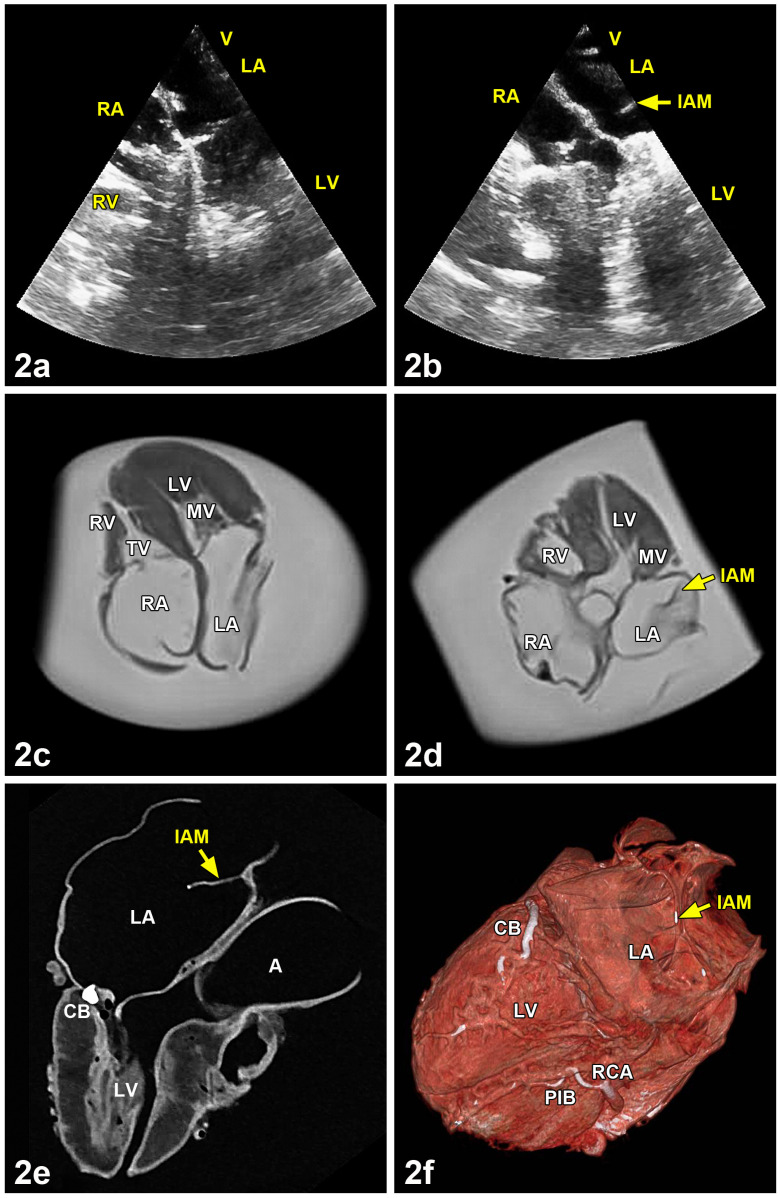
Ultrasound, magnetic resonance imaging and computed tomography analysis of a regularly formed heart in comparison with the cor triatriatum sinistrum of a 60-year-old woman. (**2a**) The regularly formed heart showed no exceptional features in the left atrium (LA). (**2b**) By contrast, the cor triatriatum presented a small intra-atrial membrane (IAM) in the left atrium (LA). (**2c**) Magnetic resonance imaging analysis of the regularly formed heart from (**2a**). No membrane was seen in the left atrium (LA). (**2d**) The left atrium (LA) of the cor triatriatum sinistrum showed a membrane (IAM) originating from the middle of the atrial septum. (**2e**) Three-dimensional reconstruction of a four-chamber view of the cor triatriatum of a 60-year-old woman by computed tomography using a gray scale image. Compared to ultrasonic and magnetic resonance imaging, this technique allowed the best visualization of the variant intra-atrial membrane (IAM) dividing the left atrium (LA) into two parts. Furthermore, it can be seen that the circumflex branch of the left coronary artery (CB) was partly mineralized. (**2f**) Three-dimensional reconstruction of a four-chamber view of the cor triatriatum sinistrum by computed tomography using virtual atriotomy. Note the topography of the intra-atrial membrane (IAM) in the left atrium (LA). In the circumflex branch of the left coronary artery (CB), in the right coronary artery (RCA) and in the posterior interventricular branch of the right coronary artery (PIB), areas of calcification were visible. A = Aorta, CB = Circumflex branch of left coronary artery, IAM = Intra-atrial membrane, LA = Left atrium, LV = Left ventricle, MV = Mitral valve, PIB = Posterior interventricular branch of the right coronary artery, RA = Right atrium, RCA = Right coronary artery, RV = Right ventricle, TV = Tricuspidal valve.

**Table 1 medicina-57-00777-t001:** Measurements in atrial and ventricular wall thickness in six male and nine female regularly formed hearts and a cor triatriatum sinistrum. M = male, F = female, Cts = Cor triatriatum sinistrum, SEM = Standard error of mean. Unit of measurements in mm.

Gender	Age	Right Atrium	Left Atrium	Right Ventricle	Left Ventricle
M	77	2	2	6	15
M	85	4	3	5	14
M	84	3	1	9	11
M	78	1	1	7	15
M	90	1	1	6	15
M	80	4	2	5	14
M	78	2	2	4	18
		Mean = 2.43	Mean = 1.71	Mean = 6.00	Mean = 14,57
		SEM = 0.48	SEM = 0.29	SEM = 0.62	SEM = 0.78
F	87	2	2	7	11
F	87	1	2	5	15
F	83	2	1	4	10
F	89	2	2	4	14
F	81	2	1	6	12
F	92	1	1	7	16
F	91	2	1	6	15
F	92	2	1	4	10
		Mean = 1.75	Mean = 1.38	Mean = 5.37	Mean = 12.88
		SEM = 0.16	SEM = 0.18	SEM = 0.46	SEM = 0.85
F	60	1	2	3	17
C.t.s.					

**Table 2 medicina-57-00777-t002:** Occurrence of variations in heart atria in a reference series of body donors.

Heart Compartment	Hearts	Variation	*n*	%
Left atrium	59	Atrial septum smooth	26	44.0
		Atrial septum with sickle-like fold, directed dorsally	20	33.9
		Atrial septum with sickle-like fold, directed cranially	1	1.7
		Atrial septum with sickle-like fold, directed caudally	1	1.7
		Atrial septum with various sieve-like depressions	4	6.8
		Oval fossa bulged in direction of left atrium	7	11.9
Right atrium	58	Oval fossa barely expressed	7	12.0
		Oval fossa flat	28	48.3
		Oval fossa deepened	23	39.7
Right atrium	59	Eustacian valve not expressed	15	25.4
		Eustacian valve normally expressed	29	49.2
		Eustacian valve strongly expressed	15	25.4
Right atrium	58	Thebesian valve not expressed	16	27.6
		Thebesian valve normally expressed	32	55.2
		Thebesian valve strongly expressed	10	17.2

## Data Availability

Not applicable.

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
