# Peer review of "Cor Triatriatum Sinistrum Combined with Changes in Atrial Septum and Right Atrium in a 60-Year-Old Woman"

_medicina, 2021, doi:10.3390/medicina57080777_

Round 1
Reviewer 1 Report
The authors present interesting case report of incomplete cor triatriatum sinistrum where the intraatrial membrane was found. The case is documented by classic anatomical dissection, ultrasonography, MRI and CT. In addition, the authors did the study of 59 regular formed hearts to focus on features in the left and right atrium. The manuscript is very carefully written. However I have couple comments that should be addressed:
1) Did authors perform dissection of the lungs? I would expect marks of pulmonary hypertension. Was there any anomaly of the pulmonary veins?
2) I miss the measurments of the atrial and ventricular wall thickness and comparison with regularly formed hearts. I would expect thicker atrial wall in cor triatriatum.
Author Response
Comments and Suggestions for Authors
The authors present interesting case report of incomplete cor triatriatum sinistrum where the intraatrial membrane was found. The case is documented by classic anatomical dissection, ultrasonography, MRI and CT. In addition, the authors did the study of 59 regular formed hearts to focus on features in the left and right atrium. The manuscript is very carefully written. However I have couple comments that should be addressed:
1) Did the authors perform dissection of the lungs? I would expect marks of pulmonary hypertension. Was there any anomaly of the pulmonary veins?
Response: Thank you for this hint. Dissection of the lungs including the pulmonary veins did not reveal any obvious pathological alterations.
2) I miss the measurments of the atrial and ventricular wall thickness and comparison with regularly formed hearts. I would expect thicker atrial wall in cor triatriatum.
Response: You are absolutely right! However, the body donors of our reference series (59 hearts) from dissection courses in the summer terms 2016, 2017 and 2018 are now interred. Therefore, atrial and ventricular wall thickness was measured in the variant heart and in regularly formed hearts of 7 males and 8 females from the dissection course held in the summer term 2021 at the Institute of Anatomy of the Rostock University Medical Center. Measurements were summarized in Table 1. Mean atrial and ventricular wall thickness on the right and left side were compared with the respective values of the cor triatriatum sinistrum. There were no thicker atrial walls either on the left or the right side in the cor triatriatum sinistrum.
Concerning the measurements, supplements were performed in „Abstract, Material and Methods, Results and Discussion“. Furthermore, an additional table was included.
Submission Date
23 June 2021
Date of this review
15 Jul 2021 12:30:19
Formularende
© 1996-2021 MDPI (Basel, Switzerland) unless otherwise stated
Reviewer 2 Report
Authors describe a cor triatriatum sinistrum combined with changes in atrial septum and right atrium in a 60-year-old woman. They conclude that their descired case may suggest that malformations in the development of the atrial septum and in the regression of the valve of the right sinus vein are involved in the pathogenesis of a cor triatriatum sinistrum.
Introduction:
Authors state that ""Therefore, the possibility of a left atrial membrane should be considered if there are signs and symptoms of mitral stenosis"" but do not provide respective References. I a cardiologists view such References are essential to confim a clinical relevance of this case.
Methods:
Authors describe ""The observed variations in the left and right atria of a reference series of 59 quite regular formed hearts are summarized in Table 1. At first, we analyzed the surface of the atrial septum seen from the left atrium. In 59 normal hearts, a smooth atrial septum was found in 44%. In the remaining 56% we observed special structures on or in the atrial septum"". Please provide more background of this "References Series".
Conclusions:
While the first part of this Section is fine for me the authors last sentence ""The cor triatriatum sinistrum of a 60-year-old female body donor described here indicates that patients with this congenital heart failure can reach an advanced age if intra-atrial communication is broad enough and there is no flow obstruction between the proximal and distal chambers of the left atrium"" should be re-written. In a cardiologists view this statment is true for several heart defects - if they are not hemodynamically relevant they do not cause significant problems during life.
Author Response
Comments and Suggestions for Authors
Authors describe a cor triatriatum sinistrum combined with changes in atrial septum and right atrium in a 60-year-old woman. They conclude that their descired case may suggest that malformations in the development of the atrial septum and in the regression of the valve of the right sinus vein are involved in the pathogenesis of a cor triatriatum sinistrum.
Introduction:
Authors state that ""Therefore, the possibility of a left atrial membrane should be considered if there are signs and symptoms of mitral stenosis"" but do not provide respective References. I a cardiologists view such References are essential to confim a clinical relevance of this case.
Response: Sorry, but I forgot the citation. Slight et al. (2004) mentioned, „that the possibility of cor triatriatum sinistrum with a left atrial membrane should be actively considered when signs and symptoms of mitral stenosis are present.“ This citation has now been added.
Methods:
Authors describe ""The observed variations in the left and right atria of a reference series of 59 quite regular formed hearts are summarized in Table 1. At first, we analyzed the surface of the atrial septum seen from the left atrium. In 59 normal hearts, a smooth atrial septum was found in 44%. In the remaining 56% we observed special structures on or in the atrial septum"". Please provide more background of this "References Series".
Response: We completed „Material and Methods“ with a more comprehensive description of the reference series of 59 quite regularly formed hearts. In detail, this series comprised 33 males between 51 and 99 years of age and 26 females between 57 and 97 years of age. The limitation of the project is that only hearts from body donors aged 51-99 years were examined for comparison and therefore age changes cannot be excluded.
Conclusions:
While the first part of this Section is fine for me the authors last sentence ""The cor triatriatum sinistrum of a 60-year-old female body donor described here indicates that patients with this congenital heart failure can reach an advanced age if intra-atrial communication is broad enough and there is no flow obstruction between the proximal and distal chambers of the left atrium"" should be re-written. In a cardiologists view this statment is true for several heart defects - if they are not hemodynamically relevant they do not cause significant problems during life.
Response: Thank you very much for this important hint. We now have rewritten this passage in the following way: „The intra-atrial communication in the cor triatriatum sinistrum of a 60-year-old female body donor described here was wide and there was no significant flow obstruction between the proximal and distal chambers of the left atrium. However, surgical treatment is indicated in patients with obstructive cor triatriatum sinistrum where the connection between two chambers is restricted in a way similar to mitral stenosis.
Submission Date
23 June 2021
Date of this review
16 Jul 2021 11:15:38
Formularende
© 1996-2021 MDPI (Basel, Switzerland) unless otherwise stated